# Analysis of carbon emission performance and regional differences in China's eight economic regions: Based on the super-efficiency SBM model and the Theil index

Yuan Zhang[1], Zhen Yu[2]*, Juan Zhang[3]

**1** School of Management, China University of Mining & Technology (Beijing), Beijing, China, **2** State Key Laboratory of Precision Measuring Technology and Instrument, Tianjin University, Tianjin, China, **3** College of Architectural Engineering, Qingdao Binhai University, Qingdao, China

* solseagull@163.com

**Data Availability Statement:** All relevant data are within the manuscript and its Supporting Information files.

**Funding:** The authors received no specific funding for this work.

## Abstract

China's carbon emission performance has significant regional heterogeneity. Identified the sources of carbon emission performance differences and the influence of various driving factors in China's eight economic regions accurately is the premise for realizing China's carbon emission reduction goals. Based on the provincial panel data from 2005 to 2017, the super-efficiency SBM model and Malmquist model are constructed in this paper to measure regional carbon emission performance's static and dynamic changes. After that, the Theil index is used to distinguish the impact of inter-regional and intra-regional differences on different regions' carbon emissions performance. Finally, by introducing the Tobit model, the effect of various driving factors on carbon emission performance differences is analyzed quantitatively. The results show that: (1) There are significant differences in different regions' carbon emission performance, but the overall carbon emission performance presents an upward fluctuation trend. Malmquist index decomposition results show substantial differences in technology progress index and technology efficiency index in different regions, leading to significant carbon emission performance differences. (2) Overall, inter-regional differences contribute the most to the overall carbon emission performance, up to more than 80%. Among them, the inter-regional and intra-regional differences in ERMRYR contributed significantly. (3) Through Tobit regression analysis, it is found that residents' living standards, urbanization level, ecological development degree, and industrial structure positively affect carbon emission performance. On the contrary, energy intensity presents an apparent negative correlation on carbon emission performance. Therefore, to improve the carbon emission performance, we should put forward targeted suggestions according to the characteristics of different regional development stages, regional carbon emission differences, and influencing driving factors.

**Competing interests:** The authors have declared that no competing interests exist.

## Introduction

Climate change is one of the most severe environmental problems the world is facing today. Both human activity and greenhouse gas emissions contribute to global climate change [1]. As one of the leading greenhouse gases, carbon dioxide is closely related to climate change. With the development of the economy, carbon dioxide emissions will continue to grow. So, the environmental problems caused by carbon dioxide emissions have attracted scholars worldwide. In the face of increasingly severe environmental issues, as the largest developing country, China's responsibility and obligation are to reduce carbon emissions and improve the ecological environment [2]. To actively respond to climate change, the Chinese government has promised to gradually reduce emission after 2030 [3]. The imbalance and disharmony of China's regional development cause substantial differences in regional carbon emission levels. Therefore, whether China's carbon emission reduction targets can be achieved successfully depends on the macro-control at the national level and the formulation of the "common but differentiated responsibilities" principle at the regional level [4].

Due to China's vast territory, different regions have significant differences in resource endowment, economic development level, urban development level, industrial structure, ecological environment, and other aspects, which lead to considerable carbon emission performance differences. Therefore, identifying the characteristics of carbon emissions in different regions accurately and discussing how to improve carbon emissions performance become the key to realizing China's carbon dioxide emission reduction target at an early date. However, if only provincial-level carbon emission policies are formulated, the decision-making cost will be increased, and it is not conducive to the national carbon trading market's unification. Therefore, some scholars try to divide China into three regions, namely the eastern, central, and western regions, to study and analyze the differences in carbon emissions of the three regions and put forward the carbon emission reduction targets for the three regions of China [5,6]. This partitioning approach is relatively rough. Therefore, according to the characteristics of China's regional economic growth and the level of economic development, this paper selects eight economic regions with a similar level of economic development determined by the development research center of the State Council of China as the research objects. The extensive analysis of the differences in carbon emission performance and the main driving factors in different regions are helpful to formulate carbon emission reduction measures suitable for different regions and promote China's overall carbon emission reduction targets. Among the research concern global environmental issues, there are more and more studies on carbon emission estimation [7], carbon emission influencing factors [8–10], carbon intensity attenuation rate [11], and carbon emission performance [12,13]. Ramanathan [14] used the DEA model to measure carbon emission performance differences among countries regarding carbon emission performance methods. Zhou et al. [15] combined the DEA model and Malmquist index to analyze the carbon emission performance of 18 countries. However, because the traditional DEA only focuses on the expected output in economic activities, it does not fully consider the unexpected output. Therefore, it is easy to deviate the measured results from the actual situation [16]. Considering unpredictable production conditions, some scholars adopt improved models to measure carbon emission performance. Du et al. [17] adopt the directional distance function model to measure China's carbon emission performance. Chang et al. [18] and Wang et al. [19] established a DEA-SBM model to measure transportation's carbon emission performance. Then, Zhang et al. [20] introduced a super efficiency SBM model to calculate each province's carbon emission efficiency, reflecting each region's carbon emission differences. Therefore, based on the previous studies, this paper introduces the total factor index for analysis, selects capital, labor, and energy consumption as input indicators, and takes

Gross Domestic Product (GDP) and carbon dioxide emission as expected output and unexpected output in economic production, respectively, to accurately measure the carbon emission performance of different regions.

Different input factors may have different effects on output. To determine the influence of different input factors on various output factors, this paper also conducts sensitivity analysis on the factors to better improve regional carbon emission performance. Sensitivity analysis is a method to quantitatively describe the importance of model input variables to output variables. According to its scope, it can be divided into local sensitivity and global sensitivity. To assess multiple input factors' sensitivity more accurately, more studies now tend to use the global sensitivity analysis method [21]. Common global sensitivity analysis methods include the qualitative Morris method, Sobol method [22,23], FAST method, quantitative Extend FAST method, and ANN-based weight analysis method [24]. The Sobol method, based on the variance decomposition principle, can be used for non-linear and non-monotonic mathematical models. Its running results are robust and reliable. It can carry out quantitative equality for the sensitivity of driving factors. So it has been widely applied in environmental modeling and non-linear models in other fields [25–30]. Therefore, this paper uses the Sobol method to study the sensitivity changes of different input factors and then uses the Monte-Carlo method simulation to confirm the influence of various input factors on the results and determine the most sensitive factors.

Unlike previous studies, this paper's main research contributions include the following three aspects: (1) This paper divides China's regions in detail and studies the regional differences of carbon emission performance from dynamic and static perspectives. The article also analyzes the global uncertainty and sensitivity. It puts forward specific measures to improve the carbon emission performance of different regions, conducive to promoting the national unified carbon trading market. (2) Calculate the size and variation trend of inter-regional and intra-regional differences in carbon emission performance of eight economic regions, which is conducive to improving carbon emission reduction targets with regional differences. (3) According to the Tobit regression model, the influencing factors of carbon emission performance values in different regions and their influencing degrees are analyzed at a deep level, conducive to putting forward targeted suggestions for improving carbon emission performance in different regions.

## Data and methodology

### Study area

This paper selects eight economic regions based on similar economic development levels determined by the State Council's development research center of China as the research object to make a more precise division in China. It is specifically divided into the following eight economic regions: Northeast Economic Region (NEER), Northern coastal economic region (NCER), Eastern coastal economic region (ECER), Southern coastal economic region (SCER), Economic region in the middle reaches of the Yellow River (ERMRYR), Economic region in the middle reaches of the Yangtze River (ERMRYTR), Southwest economic region (SWER), and Northwest economic region (NWER) (Fig 1).

Economic indicators, land use, and environmental factors vary significantly from region to region. According to the average of the data of the research period, the region with the highest economic level is ECER, which is 828005 billion yuan; the region with the lowest economic level is NWER, which is only 1,142.929 billion yuan; and the region with the highest added value of the tertiary industry is ECER, which is 3,622.941 billion yuan. The area with the most significant population density is ERMRYR, up to 14,477.85 people per square kilometer. The

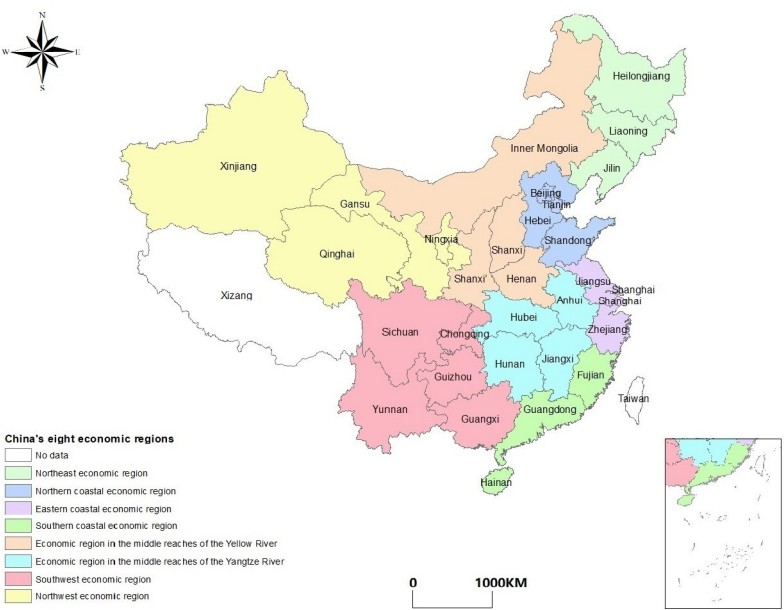

**Fig 1. The locations of China's eight economic regions.**

region with the largest afforestation area is also ERMRYR, up to 1,459.29 thousand hectares. This is related to the characteristics of the Yellow River Basin, which is caused by massive afforestation to prevent soil erosion in this region. The region with the highest water resources per capita, NWER, is much higher than other regions, closely related to the small population in this region. The region with the highest level of urbanization is SCER (Table 1). Regional resource endowments and different development stages are the fundamental reasons for various carbon emission performances.

## Data sources and data processing

According to the previous research [31], the three input indicators selected in this paper are capital stock, labor force, and energy consumption. The expected output is GDP, and the unexpected output is carbon dioxide emission. The statistical description of the primary input, expected output, and unexpected output is shown in Table 2.

Labor is expressed as urban employees, GDP, and the capital stock is calculated by select 2005 as the initial year [32]. Energy consumption and carbon dioxide emission are calculated

**Table 1. The mean value of China's eight economic regions' relevant data from 2005 to 2017.**

| Indicator (Unit) | Population density (person/square kilometer) | Added-value of tertiary industry (100 million yuan) | GDP (100 million yuan) | Urbanization rate (%) | Per capita water resources (m3/person) | Afforestation area (thousand hectares) |
|---|---|---|---|---|---|---|
| NEER | 8298.23 | 13738.04 | 35452.51 | 58.62 | 4378.75 | 367.61 |
| NCER | 7974.15 | 33244.82 | 80709.76 | 54.70 | 757.27 | 547.76 |
| ECER | 7253.31 | 36229.41 | 82800.05 | 64.86 | 2749.85 | 97.55 |
| SCER | 7425.85 | 25826.69 | 61185.65 | 62.95 | 9774.47 | 248.40 |
| ERMRYR | 14477.85 | 16848.77 | 49077.57 | 45.47 | 3618.48 | 1459.29 |
| ERMRYTR | 11929.46 | 19177.68 | 49754.78 | 46.73 | 9221.64 | 778.77 |
| SWER | 11652.15 | 17150.78 | 45447.26 | 41.54 | 15622.79 | 1414.25 |
| NWER | 11846.85 | 4473.29 | 11429.29 | 41.68 | 18038.18 | 647.53 |

**Table 2. System of regions' carbon emission performance input-output index.**

| Sorts | Indexes | Unit | Mean | Median | Standard deviation | Minimum | Maximum |
|---|---|---|---|---|---|---|---|
| **Input** | Capital stock | 100 million yuan | 29999.34 | 22033.62 | 23281.89 | 2874.32 | 105508.90 |
| | Labor | 10 thousand people | 477.89 | 414.47 | 342.41 | 18.20 | 1973.28 |
| | Energy consumption | million tons | 257.41 | 201.17 | 182.08 | 10.86 | 945.50 |
| **Expected output** | GDP | 100 million yuan | 13861.90 | 10559.43 | 12430.29 | 543.32 | 69943.16 |
| **Unexpected output** | Carbon dioxide emissions | million tons | 198.62 | 154.20 | 142.78 | 7.26 | 710.73 |

according to the Intergovernmental Panel on Climate Change (IPCC) [33]. The data of capital stock, labor, and GDP are collected from the China Statistical Yearbook (2006–2018). And the data of energy consumption and carbon dioxide emissions are derived from China Energy Statistical Yearbook (2006–2018). The specific calculation method is as follows.

**Calculation of capital stock.** According to previous research [34], this paper estimates the capital stock of different regions in different years by adopting the "perpetual inventory method," which is more popular internationally, as follows:

$$RDK_{it} = (1 - \delta_i)RDK_{i,t-1} + E_{it} \tag{1}$$

$$RDK_{i0} = \frac{E_{i1}}{\rho_i + \delta_i} \tag{2}$$

Where $RDK_{it}$, $RDK_{i,t-1}$ represent capital stock at time $t$ and $t-1$; $E_{it}$ represent the gross investment at time $t$; $RDK_{i0}$, $\delta_i$, and $\rho_i$ denote the initial capital stock, depreciation rate, and average growth rate of fixed investment with a constant price. This paper takes $\delta_i = 9.6\%$, and the geometric average method is used to obtain $\rho_i$.

**Energy consumption and carbon dioxide emissions.** This research selected eight major energy types, including coal, coke, crude oil, gasoline, kerosene, diesel oil, fuel oil, and natural gas. Energy consumption is calculated by converting each type of energy consumption into standard coal. We define carbon dioxide emissions in different regions as unexpected output. This paper sorts out eight primary energy consumption in different regions. The cumulative carbon dioxide emissions in different regions are calculated using the methods provided by IPCC. The formula is as follows:

$$C_j = \sum_{i=1}^{8} E_{ij} * K_{i1} * K_{i2} \tag{3}$$

Where $C_j$ represents the total carbon dioxide emission of region $j$, $E_{ij}$ represents the consumption of energy $i$ in region $j$. $K_1$, $K_2$ represents the standard coal conversion coefficient and carbon emission coefficient, respectively, as shown in Table 3.

## Measurement and decomposition of carbon emission performance

**Super-efficiency SBM model based on the unexpected output.** The super-efficiency DEA method proposed the concept of super-efficiency, and it is the improvement of the DEA method. This method's core idea is to exclude the decision-making units with insufficient

**Table 3. The correlation coefficient.**

| Coefficient type | Coal | Coke | Crude oil | Gasoline | Kerosene | Diesel oil | Fuel oil | Natural gas |
|---|---|---|---|---|---|---|---|---|
| $K_1$ | 0.7559 | 0.8550 | 0.5857 | 0.5538 | 0.5714 | 0.5921 | 0.6185 | 0.4483 |
| $K_2$ | 0.7143 | 0.9714 | 1.4286 | 1.4174 | 1.4174 | 1.4571 | 1.4286 | 1.3300 |

quantity and incomplete representation from the decision-making scope, observe the influence of the changes of input resources on the construction of DEA arbitrary boundary, and then get the impact of resource input on carbon emission performance. During economic production, the input of resources produces expected output and emerges unexpected output, such as $CO_2$. Tone [35] considered the unexpected output in the production process and proposed the SBM model, which is more suitable for the actual situation. Compared with the traditional DEA model, it can solve both the slack of input-output and the efficiency problem under unexpected output.

The super-efficiency DEA method has been widely used in industrial eco-efficiency [36], comprehensive energy efficiency [37,38], energy-saving, emission-reduction efficiency [39], and carbon emission performance [40,41]. Therefore, according to the previous research, this paper combines the advantages of the super-efficiency DEA model and the SBM model, constructs the super-efficiency SBM model based on unexpected output, and discusses the carbon emission performance of the eight economic regions. Assuming that the scale is constant, the input, expected output, and unexpected output can be expressed as follows: $x \in R^m$, $y^g \in R^{s_1}$, $y^b \in R^{s_2}$, The matrix $X, Y^g, Y^b$ can be defined as follow: $X = [x_1, x_2, x_3, \cdots, x_n] \in R^{m \times n}$, $Y^g = [y_1^g, y_2^g, y_3^g, \cdots, y_n^g] \in R^{s_1 \times n}$, $Y^b = [y_1^b, y_2^b, y_3^b, \cdots, y_n^b] \in R^{s_2 \times n}$. Where S1, S2, n, and m represent the expected output, unexpected output, the number of decision-making units, and the input unit, respectively. Suppose, $X>0, Y^g>0, Y^b>0$, then the $P$ means all the feasible cases:

$$P = \{(X, Y^g, Y^b), x \geq X\phi, y^g \geq Y^g\phi, y^b \leq Y^b\phi, \phi \geq 0\} \tag{4}$$

After incorporating unexpected outputs into the DMU, the SBM model can be indicated as Formula (5).

$$\rho = \min \frac{1 - \frac{1}{m}\sum_{i=1}^{m} \frac{s_i^-}{x_{i0}}}{1 + \frac{1}{S_1 + S_1}\left(\sum_{r=1}^{s_1} \frac{S_r^g}{y_{r0}^g} + \sum_{r=1}^{s_2} \frac{S_r^b}{y_{r0}^b}\right)}$$

$$\text{s.t.} \begin{cases} x_0 = X\phi + S^- \\ y_0^g = Y^g\phi - S^g \\ y_0^b = Y^b\phi - S^b \\ S^- \geq 0, \ S^g \geq 0, \ S^b \geq 0, \ \phi \geq 0 \end{cases} \tag{5}$$

Where $S = (S^-, S^g, S^b)$ is the relaxation variable of input and output, and $\rho$ is the efficiency value. Since model (5) is non-linear, for convenience of calculation, transformed model (5) is into the linear model (6) by Charnes-Cooper transformation.

$$\tau = \min t - \frac{1}{m}\sum_{i=1}^{m} \frac{s_i^-}{x_{i0}}$$

$$\text{s.t.} \begin{cases} 1 = t + \frac{1}{S_1 + S_1}\left(\sum_{r=1}^{s_1} \frac{S_r^g}{y_{r0}^g} + \sum_{r=1}^{s_2} \frac{S_r^b}{y_{r0}^b}\right) \\ x_0 t = X\mu + S^- \\ y_0^g t = Y^g \mu - S^g \\ y_0^b t = Y^b \mu - S^b \\ S^- \geq 0, \ S^g \geq 0, \ S^b \geq 0, \ \mu \geq 0, \ t \geq 0 \end{cases} \tag{6}$$

To ensure a more reasonable efficiency evaluation value, it is necessary to distinguish the decision-making units whose efficiency value is 1. Therefore, this paper selects the super-efficiency SBM model to calculate the carbon emission performance. The model expression is shown in (7), in which the objective function value $\rho^*$ is the efficiency value of the decision-making unit.

$$\rho^* = \min \frac{\frac{1}{m}\sum_{i=1}^{m}\frac{\bar{x}_i^-}{x_{i0}}}{\frac{1}{S_1+S_1}\left(\sum_{r=1}^{s_1}\frac{\bar{y}_r^g}{y_{r0}^g} + \sum_{r=1}^{s_2}\frac{\bar{y}_r^b}{y_{r0}^b}\right)}$$

$$\text{s.t.}\begin{cases} \bar{x} \geq \sum_{j=1,j\neq k}^{n}\phi_j x_j \\ \bar{y}^g \leq \sum_{j=1,j\neq k}^{n}\phi_j y_j^g \\ \bar{y}^b \geq \sum_{j=1,j\neq k}^{n}\phi_j y_j^b \\ \bar{x} \geq x_0, \bar{y}^g \leq y_0^g, \bar{y}^b \geq y_0^b, \bar{y}^g \geq 0, \phi \geq 0 \end{cases} \tag{7}$$

**Malmquist index.** The Malmquist (ML) index was also used to analyze the eight economic regions' carbon emission performance change rates. In this paper, the ML index from t to t + 1 is constructed. If the ML index is in the opening range of 0–1, carbon emission performance is reduced, while when the ML index is greater than 1, carbon emission performance is improved. Therefore, to make a dynamic analysis of carbon emission performance, this paper also decomposes the ML index into technical efficiency index (EC) and technological progress index (TC). The direction vector is defined as $g^t = y^t$-$b^t$. Thus, the calculation formula of the index of $t$−1 is as follows:

$$ML_t^{t+1} = EC_t^{t+1} + TC_t^{t+1} \tag{8}$$

$$ML_t^{t+1} = \left\{ \frac{1 + \overrightarrow{D_0^t}(x^t, y^t, b^t; y^t, -b^t)}{1 + \overrightarrow{D_0^t}(x^{t+1}, y^{t+1}, b^{t+1}; y^{t+1}, -b^{t+1})} \frac{1 + \overrightarrow{D_0^{t+1}}(x^{t+1}, y^{t+1}, b^{t+1}; y^{t+1}, -b^{t+1})}{1 + \overrightarrow{D_0^t}(x^{t+1}, y^{t+1}, b^{t+1}; y^{t+1}, -b^{t+1})} \right\}^{\frac{1}{2}} \tag{9}$$

$$EC_t^{t+1} = \frac{1 + \overrightarrow{D_0^t}(x^t, y^t, b^t; y^t, -b^t)}{1 + \overrightarrow{D_0^t}(x^{t+1}, y^{t+1}, b^{t+1}; y^{t+1}, -b^{t+1})} \tag{10}$$

$$TC_t^{t+1} = \left\{ \frac{1 + \overrightarrow{D_0^t}(x^t, y^t, b^t; y^t, -b^t)}{1 + \overrightarrow{D_0^t}(x^t, y^t, b^t; y^t, -b^t)} \frac{1 + \overrightarrow{D_0^{t+1}}(x^{t+1}, y^{t+1}, b^{t+1}; y^{t+1}, -b^{t+1})}{1 + \overrightarrow{D_0^t}(x^{t+1}, y^{t+1}, b^{t+1}; y^{t+1}, -b^{t+1})} \right\}^{\frac{1}{2}} \tag{11}$$

## Theil index decomposition method

Theil index was first proposed by Theil [42] in 1967. It is one of the essential indicators to measure regional economic differences. The advantage of the Theil index is that it can decompose the regional differences into two parts: intra-region and inter-region. This is conducive to further evaluating the contribution rate of inter-region and intra-regional differences to the overall regional differences. Most scholars use the Theil index to measure the impact of regional economic disparities, and few scholars use it to analyze the effects of regional carbon emissions performance [43]. This paper takes 30 provinces in China as the basic spatial unit, decomposes the Theil index by stages, and spoils the overall national differences into the differences among eight economic regions and the provinces in each economic region. Therefore, the results of

decomposition are as follows:

$$T = T_{BR} + T_{WR} = \sum_{i=1}^{n} \frac{C_i}{C} \ln\left(\frac{C_i/C}{Y_i/Y}\right) + \sum_{i=1}^{n} \frac{C_i}{C}\left[\sum_{j=1}^{m} \frac{C_{ij}}{C_i} \ln\left(\frac{C_{ij}/C_i}{Y_{ij}/Y_i}\right)\right] \quad (12)$$

Where $T_{BR}$ and $T_{WR}$ are the inter-regional and intra-regional differences, respectively; $i$ represents different regions, and $j$ represents the provinces in each region; $C$ denotes China's carbon emissions performance. $Y$ indicates the GDP.

## Sobol method of sensitivity analysis

It is a method to evaluate sensitivity based on variance decomposition, and the calculation steps are as follows. Suppose $\epsilon^k$ is increasing the function $f(x)$ is decomposed into the sum:

$$f(x_1, x_2, \cdots x_k) = f_0 + \sum_{i=1}^{k} f_i(x_i) + \sum_{1 \le i < j \le k} f_{i,j}(x_i, y_i) + \cdots + f_{1,2,\cdots,k}(x_1, x_2, \cdots x_k) \quad (13)$$

The decomposition formula's uniqueness has been proved, and multiple integrations can obtain all the decomposition terms. The total variance of $f(x)$ is:

$$Z = \int_{\epsilon^k} f^2(x)dx - f_0^2 \quad (14)$$

The decomposition formula can calculate the partial difference. ($1 \le i_t < \cdots < i_s \le k, s = 1,2,\cdots, k$)

$$Z_{i_1,i_2,\cdots,i_s} = \int_0^1 \cdots \int_0^1 f_{i_1,i_2,\cdots,i_s}^2(x_{i_1}, x_{i_2}, \cdots, x_{i_k})dx_{i_1}dx_{i_2}\cdots dx_{i_s} \quad (15)$$

The sensitivity coefficient can be obtained by the following formula, where, $1 \le i_t < \cdots < i_s \le k$.

$$S_{i_1,i_2,\cdots,i_s} = \frac{Z_{i_1,i_2,\cdots,i_s}}{Z} \quad (16)$$

Where $S_i$ represents the primary sensitivity index of $x_i$, which quantitatively describes the influence of $x_i$ on function $f(x)$. $S_{i_1,i_2,\cdots,i_s}$ represents the sensitivity index of order $s$ of $x_{i_1}, x_{i_2}, \cdots, x_{i_k}$, which is used to quantitatively describe the influence of the $s$ driving factors on the function $f(x)$. Therefore, for the model with $s$ influencing factors, the total sensitivity index $TS_{i_1}$ of variable $x_{i_1}$ can be expressed as:

$$TS_{i_1} = S_{i_1} + S_{i_1,i_2} + \cdots + S_{i_1,i_2,\cdots,i_s} \quad (17)$$

## Influencing factors of carbon emission performance based on Tobit model

Besides the input and output indicators mentioned above, regional carbon emission performance is also affected by many other factors. To further analyze the driving factors and influence degree of carbon emission performance, this paper takes the carbon emission performance (CMP) of the eight economic regions from 2005 to 2017 as the explained variable. It selects per capita GDP (RGDP), urbanization rate (URB), forest volume (FS), the proportion of tertiary industry (PDI), and energy intensity (EI) as the explanatory variables. This paper quantitatively analyzes the impact of residents' living standards, urban development degree, ecological development degree, industrial structure, and energy consumption level on the eight economic regions' carbon emission performance differences by building the Tobit

model. The formula of Tobit model is as follows:

$$Y_{it} = \begin{cases} \alpha_{it} + \beta_j X_{it} + \varepsilon, & \alpha + \beta X + \varepsilon > 0 \\ 0, & \alpha + \beta X + \varepsilon \leq 0 \end{cases} \tag{18}$$

Where $X_{it}$ is the explanatory variable, which indicates the indicators that affect the carbon emission performance, that is, the value of the $j$ external factor in the $t$ year. $Y_{it}$ is the explained variable, that is, the carbon emission performance value of the $i$ region in the $t$ year. β is the regression coefficient, and $\varepsilon$ is the random disturbance term. Based on not changing the relationship and nature of data, to eliminate heteroscedasticity, this paper takes the natural logarithm of the dependent variable data. The regression model of this paper is as follows:

$$\ln CMP = \alpha + \beta_1 \ln RGDP + \beta_2 \ln URB + \beta_3 \ln FS + \beta_4 \ln PDI + \beta_5 \ln EI + \varepsilon \tag{19}$$

Where the CMP RGDP, URB, FS, PDI, and EI represent the value of carbon emission performance, per capita GDP, urbanization rate, forest volume, the proportion of the tertiary industry, and energy intensity, respectively.

## Empirical results

### Static analysis results of carbon emission performance

This paper takes the unexpected output into account in the super-efficiency SBM model based on the traditional DEA model. The carbon emission performance results of the eight economic regions calculated by the super-efficiency SBM model are shown in Table 4.

**Analysis of changes in overall carbon emission performance.** It can be concluded from Table 3, the mean value of carbon emission performance in the eight economic regions during the study period was significantly different. SCER, ERMRYR, and SWER had carbon emission performances of 0.85, 0.80, and 0.81, respectively, where the average carbon emission performance was higher than the national average in 2005–2017. NWER, NCER, ECER, and NEER's average carbon emission performance is 0.79, 0.78, 0.75, and 0.72, respectively, close to the national intermediate level of carbon emission performance. The ERMRYTR is only 0.62, which is 17 percentage points lower than the national level. Therefore, there is great potential for the improvement of carbon emission performance in this region.

**The evolution characteristics of time series.** To compare the changing trend of carbon emission performance in different regions, Fig 2 is drawn. And the differences are analyzed from the overall annual change (Fig 2A) and the regional variation (Fig 2B). As shown in Fig 2A, the whole country's overall carbon emission performance shows a fluctuating upward

**Table 4. The carbon emission performance of China's eight economic regions in 2005–2017.**

| Regions | 2005 | 2006 | 2007 | 2008 | 2009 | 2010 | 2011 | 2012 | 2013 | 2014 | 2015 | 2016 | 2017 | Average |
|---------|------|------|------|------|------|------|------|------|------|------|------|------|------|---------|
| NEER | 0.50 | 0.54 | 0.59 | 0.63 | 0.67 | 0.71 | 0.74 | 0.77 | 0.78 | 0.81 | 0.83 | 0.86 | 0.91 | 0.72 |
| NCER | 0.58 | 0.61 | 0.64 | 0.67 | 0.70 | 0.73 | 0.76 | 0.79 | 0.82 | 0.87 | 0.91 | 0.97 | 1.04 | 0.78 |
| ECER | 0.60 | 0.63 | 0.66 | 0.69 | 0.72 | 0.75 | 0.76 | 0.80 | 0.76 | 0.78 | 0.82 | 0.89 | 0.93 | 0.75 |
| SCER | 0.80 | 0.76 | 0.80 | 0.81 | 0.81 | 0.83 | 0.85 | 0.86 | 0.84 | 0.86 | 0.90 | 0.93 | 0.99 | 0.85 |
| ERMRYR | 0.67 | 0.68 | 0.70 | 0.71 | 0.72 | 0.76 | 0.81 | 0.87 | 0.84 | 0.88 | 0.90 | 0.92 | 1.00 | 0.80 |
| ERMRYTR | 0.51 | 0.52 | 0.55 | 0.58 | 0.61 | 0.63 | 0.63 | 0.64 | 0.63 | 0.66 | 0.68 | 0.71 | 0.75 | 0.62 |
| SWER | 0.75 | 0.74 | 0.74 | 0.79 | 0.81 | 0.78 | 0.79 | 0.80 | 0.83 | 0.85 | 0.86 | 0.91 | 0.92 | 0.81 |
| NWER | 0.75 | 0.73 | 0.75 | 0.74 | 0.75 | 0.76 | 0.79 | 0.78 | 0.80 | 0.83 | 0.84 | 0.87 | 0.92 | 0.79 |
| Nation | 0.65 | 0.65 | 0.68 | 0.70 | 0.72 | 0.74 | 0.77 | 0.79 | 0.79 | 0.82 | 0.84 | 0.88 | 0.93 | 0.77 |

trend. The entire country's cumulative carbon emission performance had increased from 5.16 in 2005 to 7.16 in 2017. The eight economic regions' carbon emission performance has also improved to varying degrees. NEER and NCER have the most significant increase, with 82% and 79% respectively in 2017 compared with 2005. The carbon emission performance of all economic regions showed an upward trend, among which NCER showed the most massive increase, with the carbon emission performance value increased by 0.46, followed by NEER, ECER, ERMRYR, and ERMRYTR, with the performance value between 0.30 and 0.42, which are all higher than the national carbon emission efficiency growth of 0.29 (Fig 2B). However, there are three regions whose growth rate of carbon emission performance is far less than the national growth rate, namely SCER (0.19) and SWER (0.17) NWER (0.17). As shown in Table 4, regions with a slight increase in carbon emission performance show significant time series fluctuations. For example, SCER efficiency decreased from 0.8 in 2005 to 0.76 in 2006 and from 0.86 in 2012 to 0.84 in 2013. Meanwhile, SWER and NWER regions also fluctuated in the range of 0.75–0.79 and 0.73–0.78, respectively. The emergence of this fluctuating scenario is also the main reason for the slight increase in carbon emission performance.

**Analysis on the spatial pattern angle of regional decomposition.** To more intuitively reflect the spatial heterogeneity of carbon emission performance in different regions, each province's carbon emission performance in 2005, 2010, 2015, and 2017 is plotted (Fig 3). In 2005, the regions with the highest carbon emission performance were Hainan, Shanxi, Guizhou, Qinghai, and Ningxia, with carbon emission performance more significant than 1. Among them, Qinghai and Ningxia belong to the NWER. The lowest provinces were Liaoning, Jilin, Gansu, and Xinjiang, and their carbon emission performance values were no more than 0.5 (Fig 3A). Except for Shanxi, the areas with high carbon emission performance in 2010 are similar to those in 2005, but the carbon emission performance value drops to 0.9–0.99. Only Xinjiang has a carbon emission performance value of less than 0.6 (Fig 3B). The regions with the highest carbon emission performance in 2015 added to Tianjin, Shandong, Jiangsu, Guangdong, Inner Mongolia, Hunan, and Yunnan based on 2005, and the carbon emission performance of these regions was more significant than 0.9. The lower provinces are Xinjiang, Gansu, and Liaoning, whose carbon emission performance value is less than 0.8 (Fig 3C). In 2017, there are 16 provinces with carbon emission performance value greater than 1, and only Gansu is less than 0.80, while the performance values of other regions are between 0.84–0.99(Fig 3D).

Comparing the temporal and spatial evolution of each province, we can see that over time, the differences in carbon emission performance of each area are decreasing, and the carbon emission performance values are generally improved. It can be seen that the provinces with high carbon emission performance have been Ningxia and Qinghai in the NWER, Hainan in the SCER, and Guizhou in the SWER. This is due to the poor resource endowment, relatively backward economy, less energy consumption, and less carbon emission in this region, so the carbon emission performance is relatively high. On the contrary, Shanghai and Zhejiang in the ECER region are more developed in economy and energy consumption, which are also the critical provinces of energy conservation and emission reduction.

## Dynamic analysis results of carbon emission performance

Apart from analyzing the static characteristics of eight economic regions' carbon emission efficiency with the super-efficiency SBM model, this paper further explores the dynamic change characteristics of eight economic regions' carbon emission efficiency by using the ML index. The ML index and its decomposition results are shown in Fig 4.

From the time dimension analysis, the NEER's MI value decreased from 1.08 in 2005 to 1.07 in 2017, showing a downward volatility trend. From the decomposition value perspective,

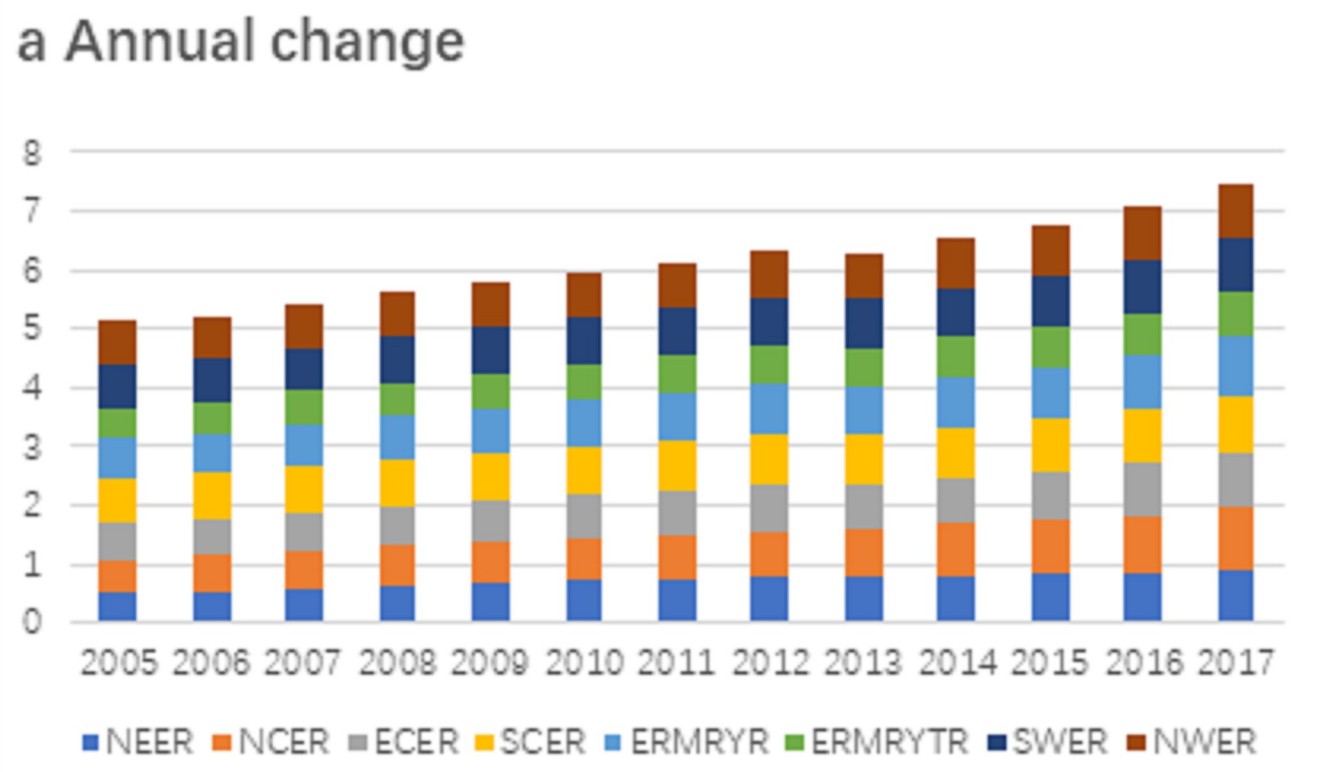

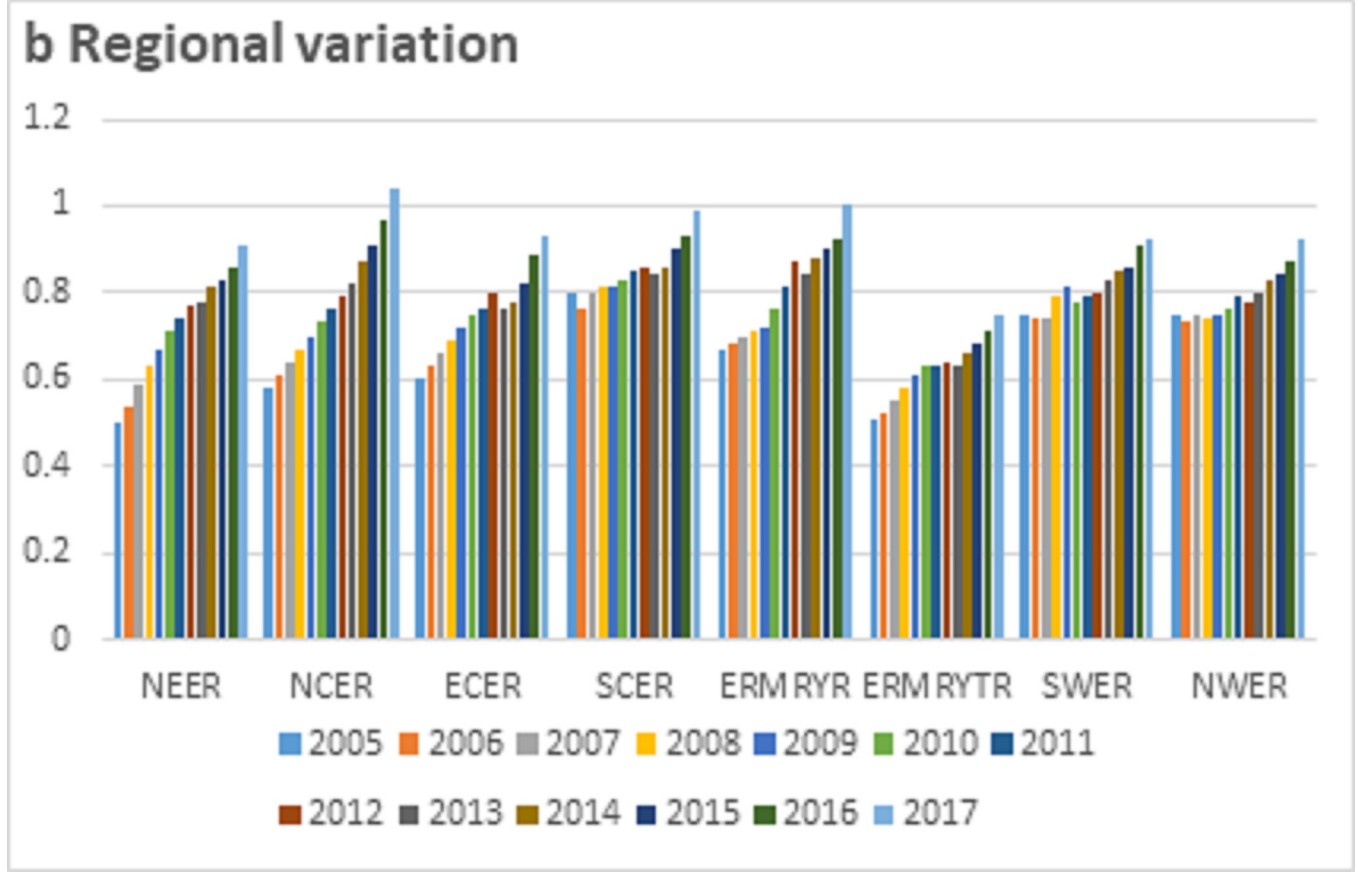

**Fig 2. Changes in carbon emission performance of China's eight economic regions during 2005–2017.**

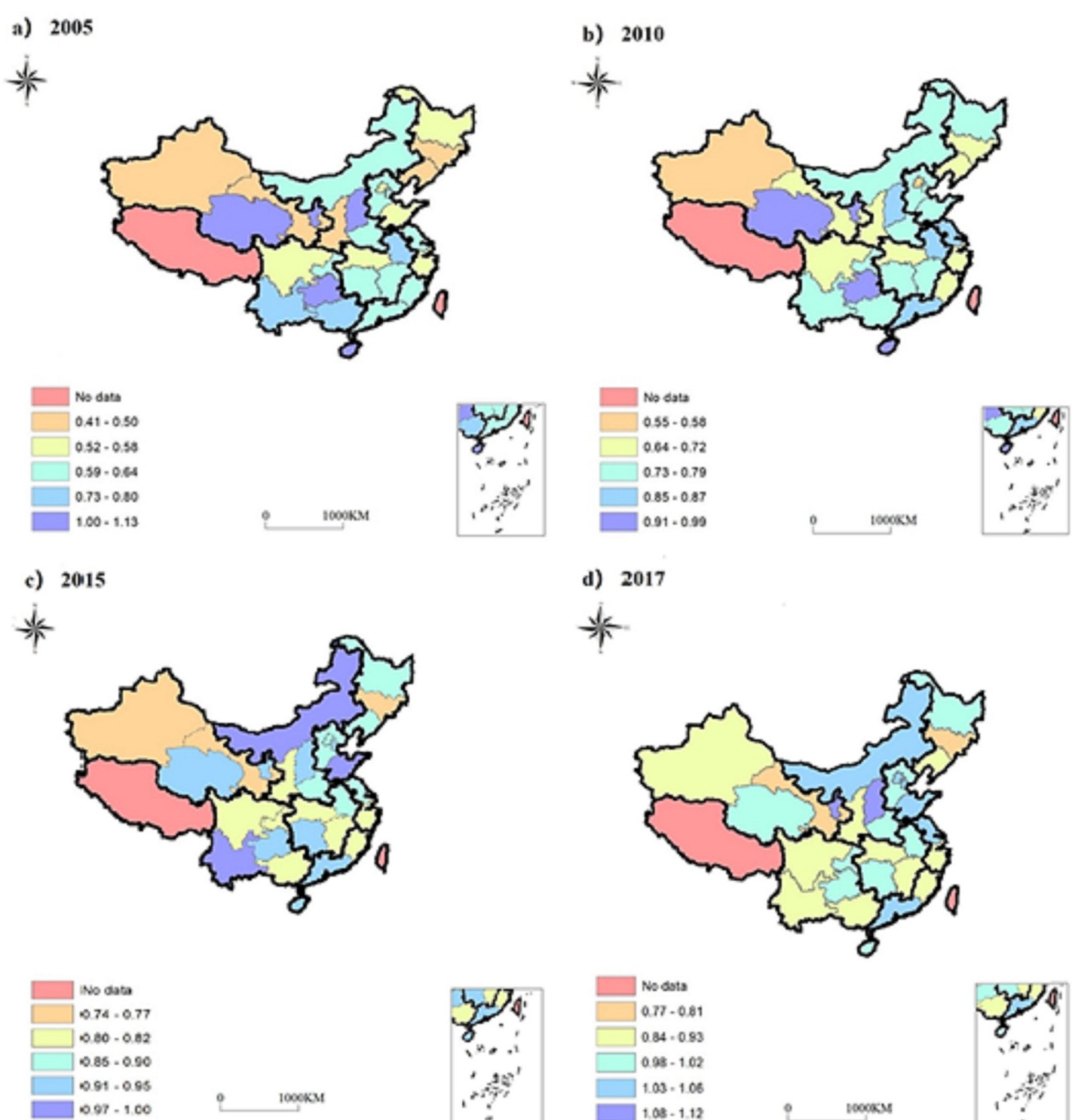

**Fig 3. The spatial pattern of carbon emission performance of various provinces from 2005 to 2017.**

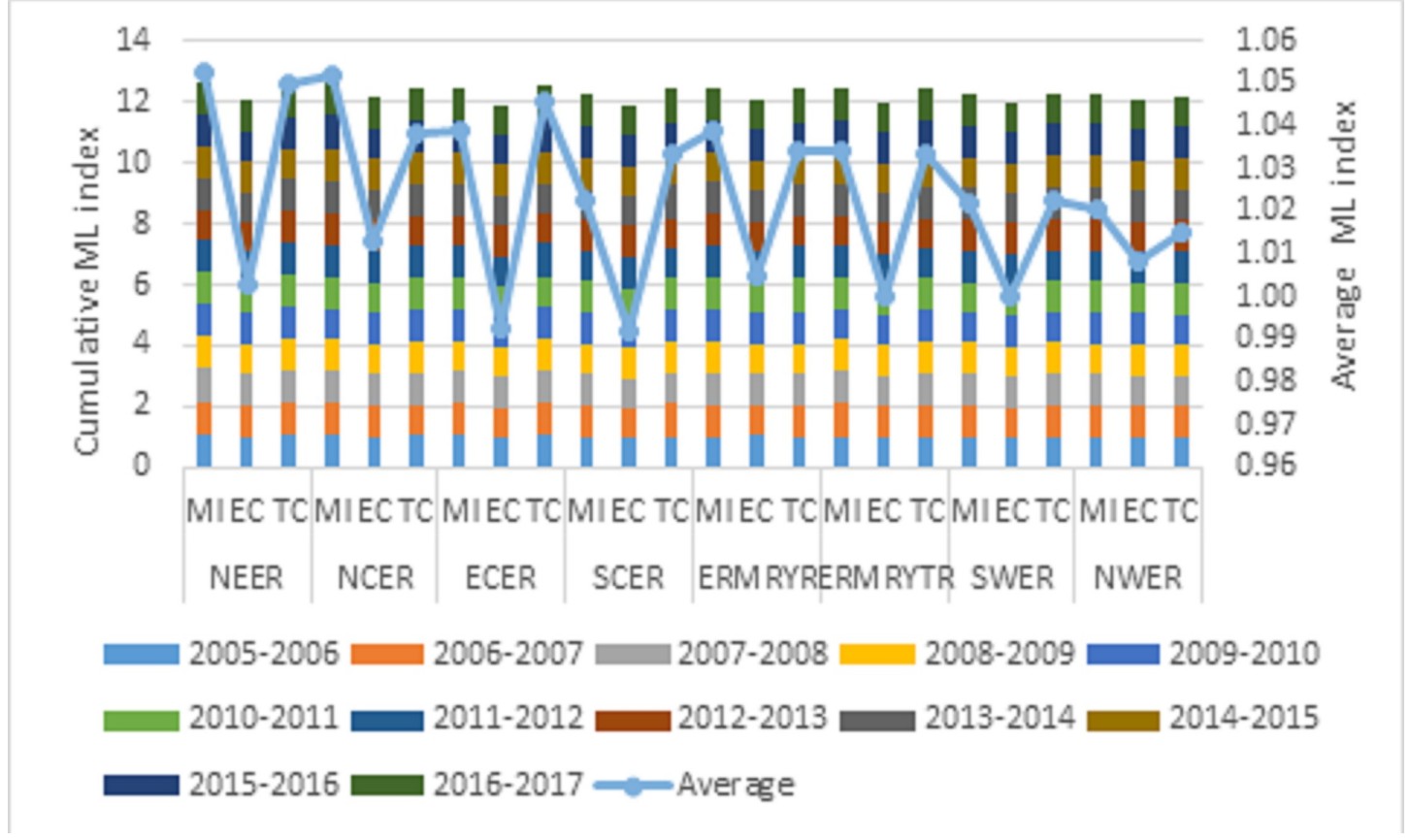

**Fig 4. The ML index and its decomposition results.**

the decrease of overall carbon emission performance in the region is mainly caused by the EC index's decline. The carbon emission performance of the other seven economic regions showed a fluctuating upward trend. The MI value of SCER increased the most, which increased by 9% from 0.98 in 2005 to 1.07 in 2017. This is mainly due to the significant improvement of the TC index in the region, which indicates that the rapid progress of technology in the region has promoted its carbon emission performance. As in SCER, the MI values of ERMRYR and ERMRYTR increased significantly, mainly due to the increase of the TC index. The other four regions have a substantial similarity, and the MI index's stable growth primarily comes from improving the technology progress index. This shows that most regions' carbon emission technology is steadily improving, but the technical efficiency index is only slightly improved. Therefore, more consideration should be given to enhancing carbon emissions' overall efficiency by improving technical efficiency. In terms of mean value, the ML index varies significantly in different regions. The MI value of NEER and NCER is the highest, reaching 1.053 and 1.052, respectively. The critical reason is that their TC index is higher in these regions than in the other regions. On the contrary, the technical efficiency index of ECER and SCER is lower than that of the other regions, only about 0.99. So, the regions should pay more attention to improving the technical efficiency to enhance the overall carbon emission performance.

### Results of the Theil index decomposition method

**Overall difference of the Theil index.** According to Eq (12), the Theil index and its decomposition value of China's eight economic regions are obtained. Overall, the inter-regional differences account for more than 80% of the fundamental differences and show a slight growth trend. The intra-regional differences only account for less than 15% of the overall contrast, slightly decreasing from 2005 to 2017 (Fig 5).

**Spatial differences of regional Theil index.** The top three inter-regional Theil index regions are ERMRYR, ECER, and NWER, and the bottom three are NCER, SWER, and ERM-RYTR. The difference between the maximum and minimum regions is significant, which indicates that the carbon emission performance of different regions varies greatly. From the average contribution rate of inter-regional differences, the ERMRYR has an enormous contribution rate, with an average contribution rate of 43.46% and a slight change from 2005 to 2017. The contribution rate of ECER, SCER, and NWER is also as high as 16.26%, 13.77%, and 14.04%, respectively, among which NWER has the fastest growth rate, rising from 7.4% to 20.66%. The contribution rate of regional difference between SCER and NWER decreases gradually by about six percentage points. NCER, SWER, and ERMRYTR showed a slight increase, but the difference contribution rate between regions was still low. The specific growth trend is shown in Fig 6A.

The regional distribution of the maximum and minimum values of carbon emission performance of the intra-regional is similar to the inter-regional. Still, the difference is relatively

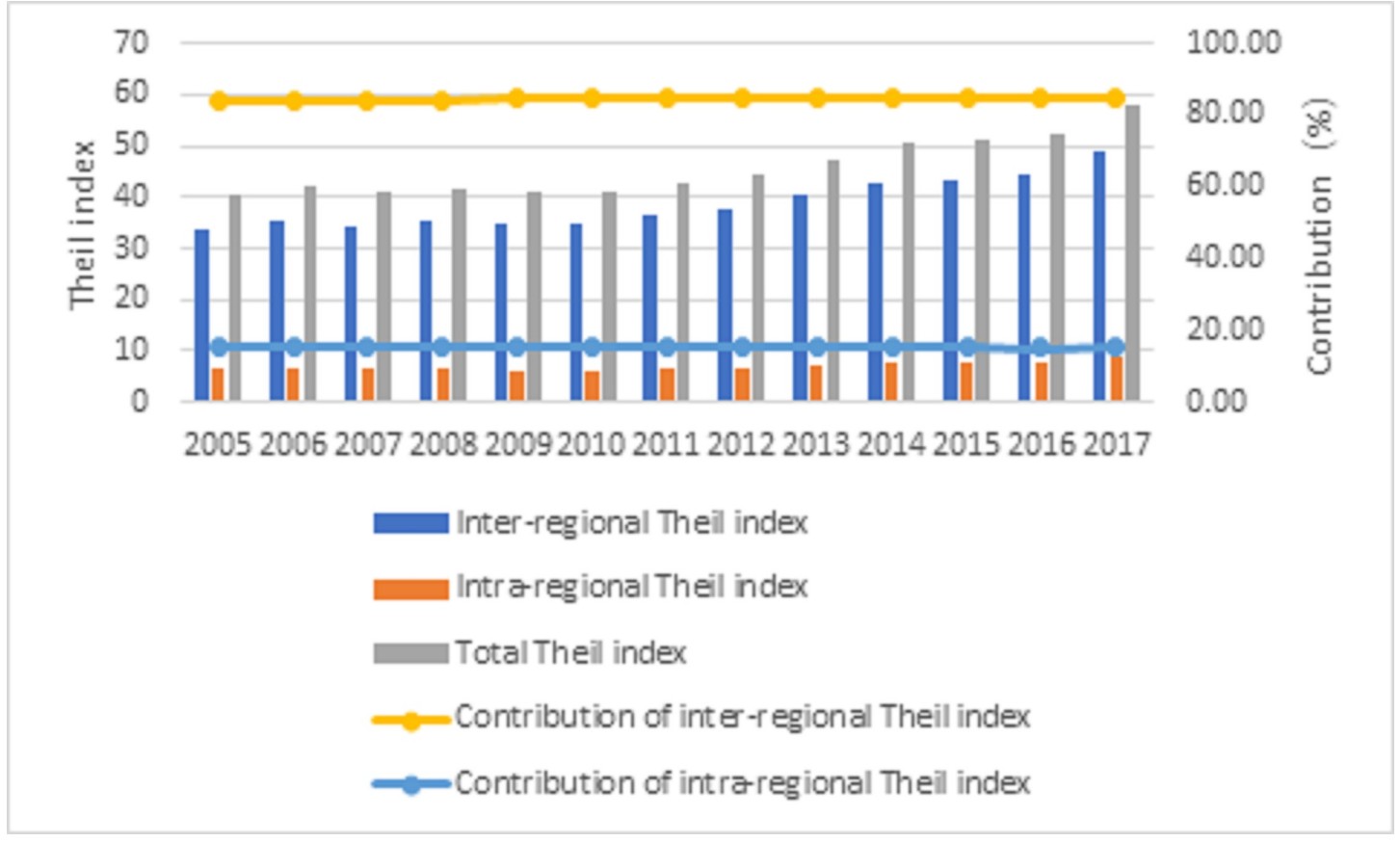

**Fig 5. Decomposition results of the Theil index in 2005–2017.**

a The inter-regional difference among the eight economic regions in 2005-2017

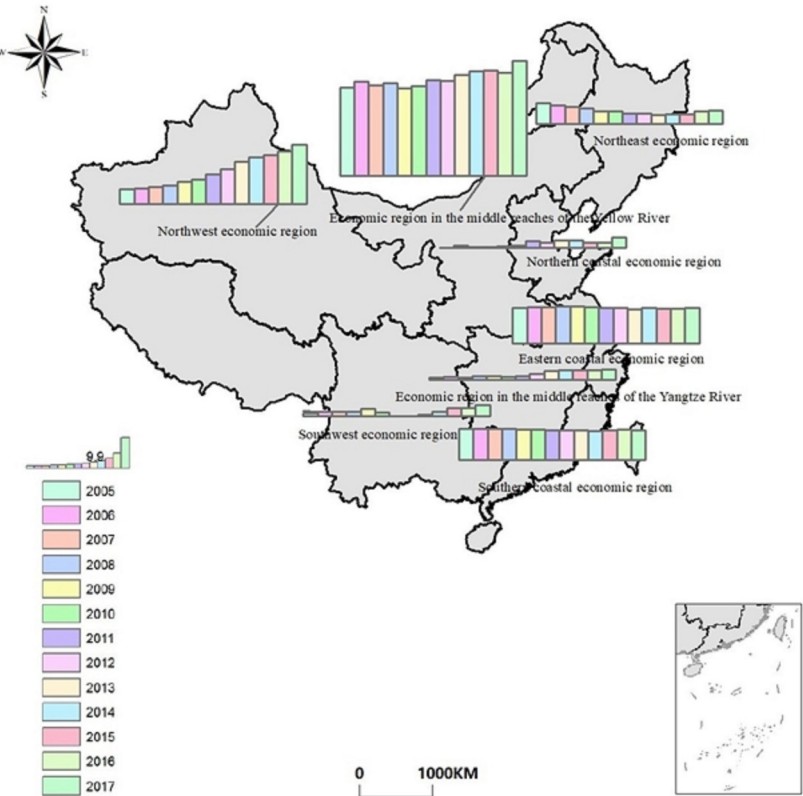

b The intra-regional difference among the eight economic regions in 2005-2017

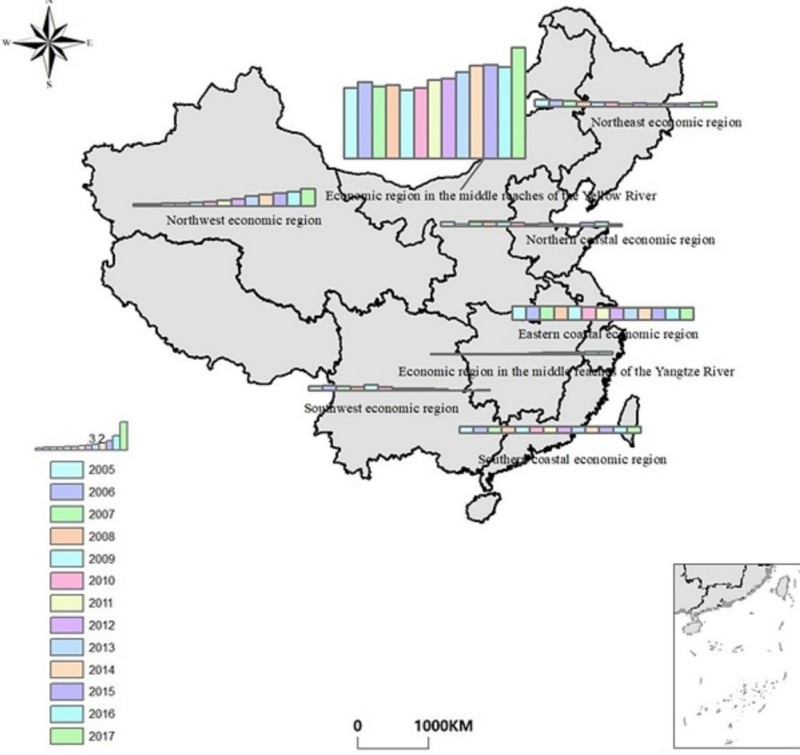

**Fig 6. Inter-regional and intra-regional differences in China's eight economic regions.**

more minor. From the average contribution rate of intra-regional differences, the contribution rate of ERMRYR is the largest, reaching up to 67.98%. From 2005 to 2017, ERMRYR showed a fluctuating growth from 64.63% to 71.69%. The second is ECER (10.56%). In this region, the contribution rate decreases rapidly with a total decrease of 4.74%, indicating that each province's carbon emission difference is getting smaller. The contribution rate of NEER, NCER, SWER, and SCER all decreased to different degrees, which indicates that each region has begun to pay attention to the adjustment of carbon emission differences to achieve regional coordinated development. The intra-regional difference contribution rate on the time series of each region is shown in Fig 6B.

## Discussion

### Structural decomposition and parameters of the Sobol method

Simulink is a visual simulation tool of MATLAB based on the block diagram design environment of MATLAB and can be used to realize dynamic system modeling, simulation, and analysis. In this paper's study, the Monte-Carlo simulation was carried out, and the SIM command of MATLAB was used to call the model. The simulation results were obtained, and the output range of the model and each driving factor was determined, which was used to analyze the source of uncertainty of the model. In the sensitivity analysis of this section, the $X_1$, $X_2$, $X_3$, $X_4$, and $X_5$ respectively represent the three input factors of labor, capital stock, energy consumption, expected output GDP, and un-expected output carbon dioxide emissions.

The first-order effects and total effects of the test functions and parameters of 1000 iterations are calculated, as shown in Fig 7. The results show that the parameter $X_5$ is the most sensitive parameter and the parameter $X_4$ is the leat sensitive parameter. In terms of total effects, similar behavior appears in Fig 7. The overall sensitivity of $X_1$ is very high, which indicates that it has significant interaction between other parameters. Conversely, the total effect of $X_2$ and $X_4$ is very close to zero. That is, there is not much interaction between them and other parameters. The results show that labor has the highest sensitivity to carbon emission performance, followed by energy consumption and carbon emissions. Therefore, from the perspective of input-output analysis, people's work efficiency can be improved by promoting high-tech industries and appropriately improving artificial intelligence technology to improve the performance of carbon emissions. Secondly, it can also encourage carbon emission performance by developing energy-saving and emission reduction technologies and implementing carbon sequestration and other technologies.

### The influencing factors of carbon emission performance

Using the Tobit model to analyze the influencing factors of carbon emission performance in eight economic regions of China, the results are shown in Table 5. Residents' living standard has a significant positive impact on carbon emission performance. The correlation coefficient is the largest, indicating that people's environmental quality requirements are also continuously improved, thus promoting regional carbon emission performance to enhance residents' living standards. The degree of urban development has a significant positive impact on carbon emission performance, which indicates that regions with high urbanization rates are more inclined to adopt new technologies in action and have higher energy utilization efficiency to achieve higher carbon emission performance. The degree of ecological development has a significant impact on carbon emissions performance at the 1% level. The correlation coefficient is high, closely related to China's commitment to achieve the peak of carbon emissions in 2030 and achieve carbon neutrality in 2060. The influence of industrial structure on carbon emission is significant at the level of 10%, indicating that with the continuous increase of the

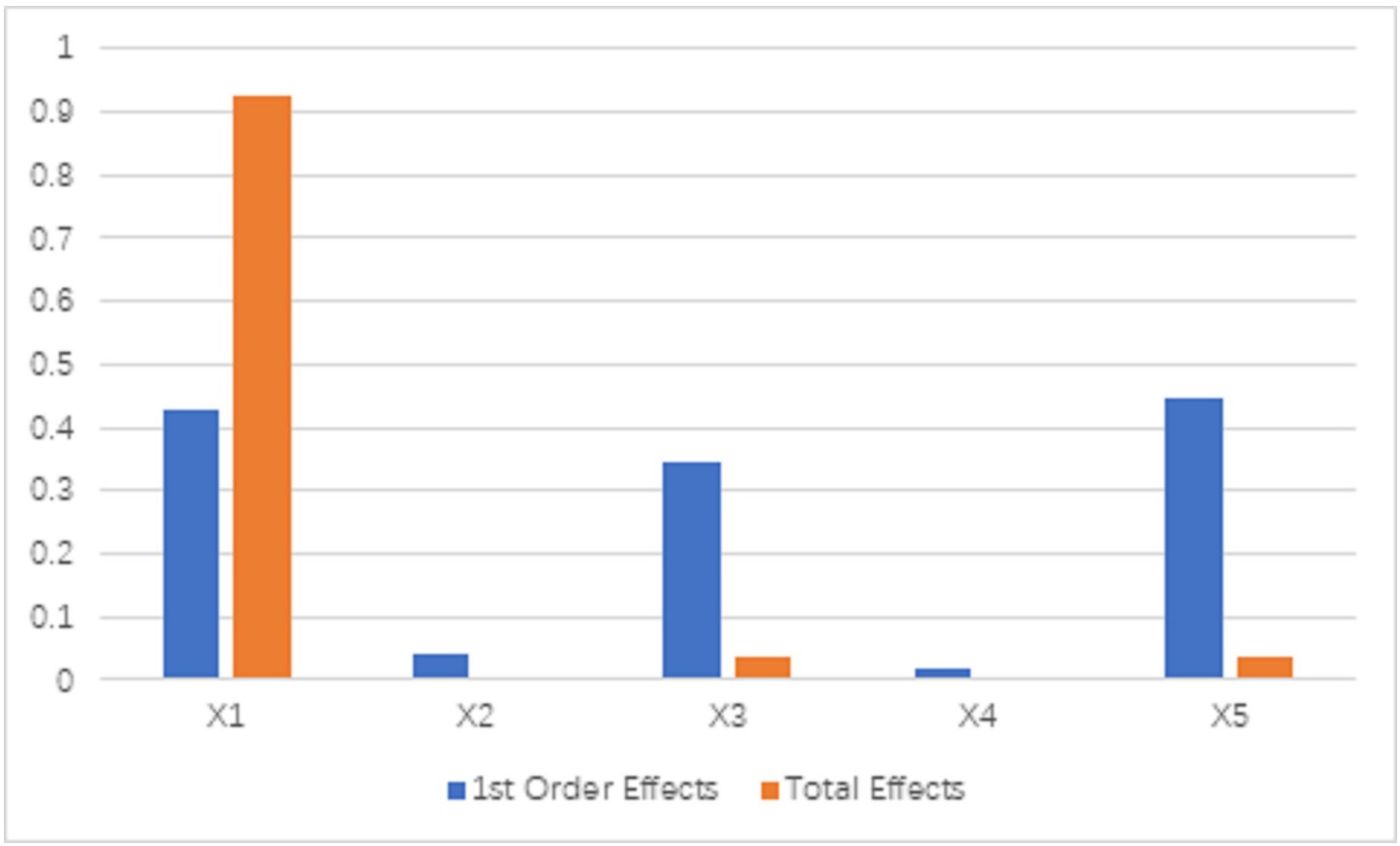

**Fig 7. First-order effects and total effects of the five parameters using Sobol's method of sensitivity analysis.**

tertiary industry proportion, carbon emission performance will be promoted. Different from other factors, energy consumption level has a significant negative impact on carbon emission performance, indicating that under the condition of a certain level of economic development, the higher the energy consumption is, the more carbon emissions will be generated, and the lower the carbon emission performance will be.

## Conclusions and limitations

### Conclusions

Based on the provincial panel data from 2005 to 2017, this paper makes an empirical analysis on the differences in carbon emission performance, the driving factors, and their influence

**Table 5. Regression results of the Tobit model.**

| Explanatory variable | Coef. | Std. Err. | t | P>|t| | 95% Conf. Interval |
|---|---|---|---|---|---|
| ln RGDP | 0.0643 | 0.0073 | 8.7900 | 0.0000 | [0.0499,0.0787] |
| ln URB | 0.0423 | 0.1935 | -0.2200 | 0.08270 | [0.0228,0.3383] |
| ln FS | 0.0404 | 0.0080 | 5.0400 | 0.0000 | [0.0246,0.0561] |
| ln PDI | 0.0255 | 0.1955 | 0.1300 | 0.08960 | [-0.3590,0.4099] |
| ln EI | -0.0167 | 0.0069 | 2.4200 | 0.0160 | [-0.0031,0.0303] |

degree in China's eight economic regions. According to the results above, some main conclusions are drawn as follows.

(1) During the study period, the eight economic regions' carbon emission performance showed significant differences. As time went on, the overall carbon emission performance showed a fluctuating upward trend. The average carbon emission performance of SCER, SWER, and ERMRYR is 0.85, 0.81, and 0.80, respectively, which is significantly higher than the national intermediate level. On the contrary, the carbon emission performance of ERMRYTR is only 0.62, which is 17% lower than the national average level, mainly due to the poor carbon emission performance of the Hubei and Jiangxi provinces in this region. Therefore, when improving the overall carbon emission performance of the ERMRYTR region, emphasis should be placed on enhancing Hubei and Jiangxi provinces' carbon emission performance.

(2) From the dynamic analysis results of carbon emission performance, we can see that each region's TC index in the time series is higher, which is the main reason for improving the ML index. However, the EC indexes of different regions are not the same. The average of the EC values of ECER and SCER is less than 1. It can be found that the EC indexes of these two regions fluctuated wildly from 2005 to 2017, which shows that the technical efficiency of these two regions has excellent potential to improve. Overall, the eight economic regions' technical progress level is relatively fast, but the technical efficiency level is relatively low. Therefore, to promote each region's carbon emission performance, each region's technical efficiency should be improved first. The maximum output can be obtained through the technical level improvement under the same input situation, and the technological efficiency can be improved, thereby improving the carbon emission efficiency [44,45].

(3) From the perspective of differences in carbon emission performance, the overall differences in China's eight economic regions' carbon emission performance show a fluctuating upward trend. The contribution rate of inter-regional difference shows a slight upward trend, while the contribution rate of intra- regional difference by a downward trend is also consistent with Liu [46]. Among them, ERMRYR has the highest contribution rate, and the contribution rates of inter-regional and intra-regional differences to the whole country are as high as 43.46% and 67.98%, respectively. As Shanxi, Shaanxi, and Inner Mongolia are resource-based regions with a high proportion of coal consumption, it is necessary to accelerate low-carbon technology in the ERMRYR region and vigorously develop new energy to control carbon emissions from the source. We should restrict the access of high energy consumption and high pollution industries in the region, maintain the total energy consumption, adhere to the principle of green mining and utilization, and strictly grasp the environmental protection standards of traditional energy such as coal to realize its efficient transformation and the transformation of new and old kinetic energy.

(4) Through the analysis of the driving factors affecting carbon emission performance, it is shown that residents' living standard, urbanization level, ecological development degree, and industrial structure upgrading all have a significant positive impact on the improvement of carbon emission performance on the contrary, energy consumption level harms the progress of carbon emission performance [47]. Therefore, it is essential to enhance residents' living standards, promote the urbanization rate, and reduce the energy intensity to improve carbon emission performance. Meanwhile, efforts should be made to build a low-carbon economic development model, optimize the industrial structure, encourage high-tech industries, and promote the harmonious development of energy, economy, and environment.

## Strengths and limitations

In this paper, the research objects are divided into eight economic regions in China, changing the research direction of provincial or industrial level in the previous carbon emission

performance measurement. The research results are more targeted and more conducive to the unification of the national carbon trading market. First of all, the combination of the super-efficiency SBM model and Malmquist index model illustrates the characteristics of carbon emission performance from both static and dynamic perspectives, which makes up for the shortcomings of preliminary discussion of a single model. Secondly, through the decomposition of regional carbon emission performance differences, the inter-regional and intra-regional differences of carbon emission performance can be obtained, which provides convenience for reducing regional differences.

However, there are still some shortcomings in this paper. This paper only considers eight primary energy consumption to estimate the carbon emissions, which has a particular gap with the actual regional carbon emissions. In the analysis of driving factors, this paper only considers the influence of five major factors on regional carbon emission performance, without in-depth study on the impact of scientific and technological progress, government macro-regulation, average rainfall, and population aging carbon emission. We will continue to pay attention to the development of regional carbon emission performance in future work. We will improve the above deficiencies to obtain a more accurate carbon emission performance value and provide suggestions for improving regional carbon emission performance.

## Supporting information

**S1 Fig.**
(TIF)

**S2 Fig.**
(TIF)

**S3 Fig.**
(TIF)

**S4 Fig.**
(TIF)

**S5 Fig.**
(TIF)

**S6 Fig.**
(TIF)

**S7 Fig.**
(TIF)

## Author Contributions

**Conceptualization:** Juan Zhang.

**Methodology:** Zhen Yu.

**Writing – original draft:** Yuan Zhang.

**Writing – review & editing:** Juan Zhang.

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
