## [Decision Letter · Decision Letter 0]

24 Feb 2021

PONE-D-21-02089

Analysis of carbon emission performance and regional differences in China's eight economic regions：Based on the super-efficiency SBM model and the Theil Index

PLOS ONE

Dear Dr. Yu,

Thank you for submitting your manuscript to PLOS ONE. After careful consideration, we feel that it has merit but does not fully meet PLOS ONE’s publication criteria as it currently stands. Therefore, we invite you to submit a revised version of the manuscript that addresses the points raised during the review process. Please consider following suggestions during review:

Please arrange keywords in alphabetic order.

What is specific reason to select study time from 2005 to 2017? how about making it till 2020?

Introduction is too less, add some rationale and relevance to the recent studies. Reviewers have given some serious comments about all sections as well.

In doing so, please try to relate your literature to this journal to make your case strong.

Font styles are non-similar in the text and in tables/figures.

Moreover, pay attention to the equations and SI units. For instance, equation 7 seems scattered.

Tables need to edit and fit into the draft as per the journal's guidelines.

Last but not least, please pay serious attention to the reviewers' comments.

We look forward to receiving your revised manuscript.

Kind regards,

Ghaffar Ali, PhD

Academic Editor

PLOS ONE

Journal Requirements:

3.  We note that Figure 2 in your submission contain map images which may be copyrighted. All PLOS content is published under the Creative Commons Attribution License (CC BY 4.0), which means that the manuscript, images, and Supporting Information files will be freely available online, and any third party is permitted to access, download, copy, distribute, and use these materials in any way, even commercially, with proper attribution. For these reasons, we cannot publish previously copyrighted maps or satellite images created using proprietary data, such as Google software (Google Maps, Street View, and Earth). For more information, see our copyright guidelines: http://journals.plos.org/plosone/s/licenses-and-copyright.

3.1.    You may seek permission from the original copyright holder of Figure 2 to publish the content specifically under the CC BY 4.0 license. 

3.2.    If you are unable to obtain permission from the original copyright holder to publish these figures under the CC BY 4.0 license or if the copyright holder’s requirements are incompatible with the CC BY 4.0 license, please either i) remove the figure or ii) supply a replacement figure that complies with the CC BY 4.0 license. Please check copyright information on all replacement figures and update the figure caption with source information. If applicable, please specify in the figure caption text when a figure is similar but not identical to the original image and is therefore for illustrative purposes only.

Reviewers' comments:

Reviewer's Responses to Questions

**Comments to the Author**

1. Is the manuscript technically sound, and do the data support the conclusions?

Reviewer #1: Yes

Reviewer #2: Yes

2. Has the statistical analysis been performed appropriately and rigorously? 

Reviewer #1: Yes

Reviewer #2: Yes

3. Have the authors made all data underlying the findings in their manuscript fully available?

Reviewer #1: Yes

Reviewer #2: Yes

4. Is the manuscript presented in an intelligible fashion and written in standard English?

Reviewer #1: Yes

Reviewer #2: Yes

5. Review Comments to the Author

Reviewer #1: This study presents a comparative assessment of emissions in the eight economic regions in China by applying two methods, the SBM and Theil Index.

It is difficult to understand the gist of the article through the current abstract. It is strongly advised to revise the abstract considering background, introduction, objectives, methods results and inferences from the study.

Line 50: What is meant by “The Chinese government has promised to achieve the peak of carbon dioxide emissions by 2030 [5]”.

A map could have been added to indicate the locations of the eight economic regions.

Line 107: “C” in the “climate change” should be capitalized.

Authors may consider replacing the radar graph with a bar graph (maybe a stacked bar graph could better present the significant changes”.

The (1), (2), (3) could be replaced with a heading indicating a summary of the point.

CO2 should be CO2

Fig 2. Could be superimposed with the boundaries of the economic regions.

Fig 3 Intra-region change could have been plotted on the secondary y-axis.

Fig 4. The names in the legend should be in full forms to make the figure self-explanatory.

Discussion lacks the significant contribution of the study to existing literature and incorporation of references from similar studies in China and from elsewhere in the world. And it should be separative from the conclusion while the conclusion should focus on key inferences from the study.

Reviewer #2: Suggestions and comments for authors:

The paper is incredibly interesting and it connects extremely well models for a very important problem. Great job!

The only thing that is missing, unfortunately as many modeling studies, is a Global Sensitivity and Uncertainty Analysis. This can be done by apportioning the uncertainty of change to drivers (input factors of the model) and in particular to describe changes in predictands (individually or put together as a systemic indicator, see e.g. Servadio and Convertino (2018)). See e.g. Saltelli et al (2004) or M.L.Chu-Agor et al (2011) for an extensive discussion about this topic and how data should be used for GSUA using a variance-based approach that is non-linear. It is simple because you already have all data and calculations. I believe this is really important and can be done in space too to identify drivers' importance in space.

Ref:

1. Saltelli A, Marco Ratto, Terry Andres, Francesca Campolongo, Jessica Cariboni, Debora Gatelli, Michaela Saisana, Stefano Tarantola, 2004

Global Sensitivity Analysis: The Primer

ISBN: 978-0-470-05997-5

2. Optimal information networks: Application for data-driven integrated health in populations

Joseph L. Servadio1 and Matteo Convertino2,3,4,*

Science Advances 02 Feb 2018:

Vol. 4, no. 2, e1701088

DOI: 10.1126/sciadv.1701088

3. Exploring vulnerability of coastal habitats to sea level rise through global sensitivity and uncertainty analyses

M.L.Chu-Agor et al

Environmental Modelling & Software

Volume 26, Issue 5, May 2011, Pages 593-604

4. Packages for GSUA

- https://www.safetoolbox.info/info-and-documentation/

- or the original one https://cran.r-project.org/web/packages/sensitivity/sensitivity.pdf

Other items to be considered are:

1. The introduction section should be improved further by precisely incorporating the background, significance, research gaps in terms of methodology and problem statements, the contribution of this study in terms of minimizing the research gaps, specific objectives, and novelty of the research.

2. In methodology, authors are recommended to create a study area map including the eight economic zones to let the readers get an idea about the study area.

3. It is also important to present some of the socio-economic and land use, and environmental factor data such as per capita GDP, population density, forest area, green space, rate of urbanization, and average rainfall at a regional scale to compare the differences among the selected economic zones and compare with the study results.

4. The author could reconsider the presentation of results in tabular form. It would better if the author makes some maps like Figure 2 instead of Table 4,5, & 6 for better understanding at a spatial scale. In that case, the author can replace the tables in the supplementary information section.

5. The study failed to explore the influencing factors of regional carbon emission performance. This needs to be done, as many studies are available in the literature, and relevant data are also readily available in China as per the reviewer's understanding.

6. It would be great if the author adds some discussion on the reasons for spatial heterogeneity in terms of carbon emission performance in the respective sections.

7. The author can also add some literature in the policy implication section especially some of the suggestions highlighted in this study that might have connections with earlier studies of similar research fields, which will improve the weightage of the suggestions being recommended in this study.

8. The authors didn’t mention any limitations of the study, however, several limitations exist in the study. So, the authors are recommended to mention the potential limitation of the study in the discusioon or conclusion section.

6. PLOS authors have the option to publish the peer review history of their article (what does this mean?). If published, this will include your full peer review and any attached files.

Reviewer #1: No

Reviewer #2: No

---

## [Author Response · Author response to Decision Letter 0]

29 Mar 2021

Response to Reviewers' Comments

Dear Editor and Reviewers, 

Thank you for your letter and the reviewers' comments concerning our manuscript entitled "Analysis of carbon emission performance and regional differences in China's eight economic regions：Based on the super-efficiency SBM model and the Theil index" (Manuscript ID: PONE-D-21-02089). 

The editor provided suitable suggestions, and the reviewers have made wide-ranging and detailed comments on the article. The concerns and comments are all valuable and very helpful for revising and improving our manuscript. We have tried our best to improve the manuscript according to the comments and made some changes in the manuscript. 

In addition, we have carefully and individually responded to each of the reviewer's comments. The revised portion is marked in red in the revised manuscript. The main corrections in the paper and the responses to the reviewers' comments are presented in the following text, and please have your check. We hope that the revised manuscript will be approved. 

Looking forward to hearing from you soon. 

Yours sincerely, 

Zhen Yu, Ph.D. 

State Key Laboratory of Precision Measuring Technology and Instrument, Tianjin University, China 

E-mail: solseagull@163.com (ZY)

Response to academic Editor

1.Please arrange keywords in alphabetic order.

Response: Thank you very much for your suggestions. 

The keywords have been arranged in alphabetic order. 

The keywords have been revised as follows: Carbon emissions performance; China's eight economic regions; Malmquist index; Super- efficiency SBM model; Theil index

Thank you.

2.What is specific reason to select study time from 2005 to 2017? how about making it till 2020?

Response: Thank you very much for your suggestions. 

At the Paris climate conference in 2015, the Chinese government proposed to decrease the country's carbon intensity by 60%-65% by 2030 compared to the level of 2005. Therefore, this paper selects 2005 as the starting year. 

But according to the official data released by the National Bureau of Statistics of China (Official website: https://data.stats.gov.cn/index.htm), the latest relevant data of primary energy consumption data and capital stock of each province are only updated to 2017. 

 Your proposal to study the year 2020 is of great research significance and can better explain the differences and sources of China's regional carbon emission performance. Therefore, we will further study it after the official data of various provinces from 2018 to 2020 are released. Thank you for your valuable advice.

Thank you very much!

3.Introduction is too less. Add some rationale and relevance to the recent studies. Reviewers have given some serious comments about all sections as well.

Response: Thank you very much for your suggestions. 

The introduction of this paper has been modified according to experts' opinions, and some relevant research literature has been added. The specific amendments are shown in the "Introduction" of the original text and are listed in reply to the reviewers below. 

Thank you for your valuable advice.

4.Moreover, pay attention to the equations and SI units. For instance, equation 7 seems scattered.

Tables need to edit and fit into the draft as per the journal's guidelines.

Response: Thank you very much for your suggestions. 

This paper has been modified according to the journal template's requirements, such as equation, SI unit, etc. 

Thank you very much!

Response to Journal Requirements:

1.Please ensure that your manuscript meets PLOS ONE's style requirements, including those for file naming.

Response: Thank you very much for your suggestions. 

This article has been revised according to PLOS ONE's style requirements. 

2.We suggest you thoroughly copyedit your manuscript for language usage, spelling, and grammar. If you do not know anyone who can help you do this, you may wish to consider employing a professional, scientific editing service. 

Response: Thank you very much for your suggestions. 

Based on the suggestions, we have checked the entire manuscript line-by-line carefully for English usage and grammar and improved language fluency and conciseness. The relevant modification has been highlighted in the revised manuscript.

With many friends at home and abroad, this article has been revised and polished many times. Dingfei Jie (School of Management, China University of Mining and Technology Beijing, Beijing, China) is most appreciated. She is a colleague in the same field and has studied abroad for many years. She has published the paper "The future of coal supply in China based on non-fossil energy development and carbon price strategies" in the journal of Energy. At the same time, thanks to Fei Guo (International Institute for applied systems analysis, Australia), who has given many grammar rework problems. 

3.We note that Figure 2 in your submission contain map images which may be copyrighted.

Response: Thank you very much for your suggestions. 

In this paper, as shown in Figure 2, this kind of map is downloaded from the natural earth website and drawn according to the paper's needs. The essential layers of the geographic map used in my article are all obtained from the natural earth website. Pictures can be used on the Natural Earth website (Website: http://www.naturalearthdata.com/about/terms-of-use/).

The website has made a detailed description：All versions of Natural Earth raster + vector map data found on this website are in the public domain. You may use the maps in any manner, including modifying the content and design, electronic dissemination, and offset printing. The primary authors, Tom Patterson and Nathaniel Vaughn Kelso, and all other contributors renounce all financial claims to the maps and invites you to use them for personal, educational, and commercial purposes. No permission is needed to use Natural Earth.

Response to Reviewer #1:

1.It is difficult to understand the gist of the article through the current abstract. It is strongly advised to revise the abstract considering background, introduction, objectives, methods, results, and inferences from the study.

Response: Thank you very much for your suggestions. 

The abstract section of this paper has been re-edited according to the order of research background, introduction, objectives, methods, results, and inference. The newly added part is shown in red font.

The revised summary is as follows:

China's carbon emission performance has significant regional heterogeneity. Accurately identifying the sources of carbon emission performance differences and the influence degree of various driving factors in China's eight economic regions is the premise for realizing China's carbon emission reduction goals. Based on the provincial panel data from 2005 to 2017, this paper constructs the super-efficiency SBM model and Malmquist model to measure regional carbon emission performance's static and dynamic changes. Secondly, the Theil index is used to distinguish the impact of inter-regional and intra-regional differences on different regions' carbon emissions performance. Finally, by introducing the Tobit model, this paper quantitatively analyzes the impact of various driving factors on carbon emission performance differences. The results show that :(1) There are significant differences in the carbon emission performance of different regions, but the overall carbon emission performance presents a fluctuating upward trend. Malmquist index decomposition results show substantial differences in technology progress index and technology efficiency index in different regions, leading to significant carbon emission performance differences. (2) On the whole, inter-regional differences contribute the most to the overall carbon emission performance, up to more than 80%. Among them, the inter-regional and intra-regional differences in ERMRYR contributed significantly.(3) Through Tobit regression analysis, it is found that residents' living standards, urbanization level, ecological development degree, and industrial structure all have different positive effects on carbon emission performance. On the contrary, energy intensity presents an apparent negative correlation. Therefore, to improve the carbon emission performance, we should put forward targeted suggestions according to the characteristics of different regional development stages, regional carbon emission differences, and influencing driving factors.

2.Line 50: What is meant by "The Chinese government has promised to achieve the peak of carbon dioxide emissions by 2030 [5]".

Response: Thank you very much for your suggestions. 

 At the Paris climate conference in 2015, the Chinese government proposed to decrease the country's carbon intensity by 60%-65% by 2030 compared to the level of 2005. The carbon dioxide emissions will reach the highest value by 2030, and China's carbon dioxide emissions will begin to decrease after 2030. The peak of China's carbon emissions in 2030 is a strategic goal for China to deal with climate change.

3.A map could have been added to indicate the locations of the eight economic regions.

Line 107: "C" in the "climate change" should be capitalized.

Response: Thank you very much for your suggestions. 

 A map to indicate the eight economic regions' locations has been added, as shown in the revised paper Fig1.

The "C" in line 107 has been capitalized，it is modified as "Intergovernmental Panel on Climate Change (IPCC)."

Thank you very much.

4. Authors may consider replacing the radar graph with a bar graph (maybe a stacked bar graph could better present the significant changes".

Response: Thank you very much for your suggestions. 

The original radar graph (original paper: Fig 1) has been replaced by a stacked bar graph (revised paper: Fig 2).

Original paper: Fig 1

After modification: Fig 2（Revised paper）

5.The (1), (2), (3) could be replaced with a heading indicating a summary of the point.

 CO2 should be CO2.

Response: Thank you very much for your suggestions. 

"(1), (2), (3) " in Section 4.1 of the original paper has been replaced with a heading indicating a summary of the point. The content of the original paper： "(1) From the perspective of spatial pattern distribution……", "（2）According to the evolution characteristics of time series……", "（3）From the perspective of the decomposition of the provinces in each region……" Replaced by: "Analysis of changes in overall carbon emission performance," "The evolution characteristics of time series," "Analysis on the spatial pattern angle of regional decomposition."

The spelling of "CO2" in the text has been corrected to "CO2".

6. Fig 2. It could be superimposed with the boundaries of the economic regions.

 Fig 3 Intra-region change could have been plotted on the secondary y-axis.

 Fig 4. The names in the legend should be in full forms to make the figure self-explanatory.

Response: Thank you very much for your suggestions. 

（a）Modifications to the original Fig 2. 

The boundaries of the economic regions in the original Fig 2 has been added, as shown in the modified paper Fig3. The revised Fig3 is as follows. Fig 3. The spatial pattern of carbon emission performance of various provinces from 2005 to 2017.

（b）Modifications to the original Fig 3.

Some adjustments have been made to the original Fig3. To more intuitively reflect the degree of change, the contribution rate of inter-regional and intra-regional differences is plotted on the secondary y-axis, as shown in the revised paper ：Fig 5. Decomposition results of the Theil index in 2005-2017.

（C）Fig 4. The names in the legend should be in full forms to make the figure self-explanatory.

The original Fig 4has been replaced with Fig 6, and the corresponding area names have been marked in the figure. The specific changes are shown in the figures below.

The original Fig 4 The contribution rate of inter-regional difference among the eight economic regions in 2005-2017.

Revised Fig 6. Inter-regional and intra-regional differences in China's eight economic regions.

7.Discussion lacks the significant contribution of the study to existing literature and incorporation of references from similar studies in China and from elsewhere in the world. And it should be separative from the conclusion while the conclusion should focus on key inferences from the study.

Response: Thank you very much for your suggestions. 

According to the whole paper's adjustment, the discussion and conclusion have been re-edited based on the existing important literature. Specific adjustments are made to the "Discussion" and "Conclusion" parts of the revised paper.

Response to Reviewer #2:

The only thing that is missing, unfortunately as many modeling studies, is a Global Sensitivity and Uncertainty Analysis. This can be done by apportioning the uncertainty of change to drivers (input factors of the model) and in particular to describe changes in predictands (individually or put together as a systemic indicator, see e.g. Servadio and Convertino (2018)). See e.g. Saltelli et al (2004) or M.L.Chu-Agor et al (2011) for an extensive discussion about this topic and how data should be used for GSUA using a variance-based approach that is non-linear. It is simple because you already have all data and calculations. I believe this is really important and can be done in space too to identify drivers' importance in space.

Response: Thank you very much for your suggestions. 

According to the suggestions of experts, in order to find out the degree of input factors to output factors, this paper adds sensitivity analysis to each factor, in order to better improve the regional carbon emission performance. In this paper, Sobol method is used to study the sensitivity changes of different input factors, and Monte Carlo method is used to simulate and determine the influence degree of different input factors on the results, so as to find out the most sensitive factor. The results of sensitivity analysis are analyzed in the "Discussion" part of the revised paper.

Other items to be considered are:

1.The introduction section should be improved further by precisely incorporating the background, significance, research gaps in terms of methodology and problem statements, the contribution of this study in terms of minimizing the research gaps, specific objectives, and novelty of the research.

Response: Thank you very much for your suggestions. 

The introduction section has been rewritten. According to the suggestions of the experts, this paper re-integrates the research background, significance, methods and other contents mentioned above. The re- edited section is shown below:

Therefore, based on previous studies, this paper introduces the total factor index for analysis, selects capital, labor, and energy consumption as input indicators, and takes Gross Domestic Product (GDP) and carbon dioxide emission as expected output and unexpected output in economic production, respectively, to accurately measure the carbon emission performance of different regions.

Due to different input factors may have different effects on output, in order to find out the influence of input factors on output factors respectively, this paper also conducts sensitivity analysis on the factors, so as to better improve regional carbon emission performance. Sensitivity analysis is a method to quantitatively describe the importance of model input variables to output variables. According to its scope, it can be divided into local sensitivity and global sensitivity. In order to assess the sensitivity of multiple input factors more accurately, more studies now tend to use global sensitivity analysis method [21]. At present, common global sensitivity analysis methods include qualitative Morris method, Sobol method [22,23], FAST method, quantitative Extend FAST method and ANN based weight analysis method [24]. Among them, Sobol method, based on the variance decomposition principle, can be used for nonlinear and non-monotonic mathematical models. Its running results are robust and reliable, and it can carry out quantitative equality for the sensitivity of driving factors, so it has been widely applied in environmental modeling and nonlinear models in other fields [25-30]. Therefore, this paper uses the Sobol method to study the sensitivity changes of different input factors and then uses the Monte-Carlo method simulation to confirm the influence degree of different input factors on the results and find out the most sensitive factors.

Unlike previous studies, this paper's main research contributions may include the following three aspects: (1 This paper divides China's regions in detail and studies the regional differences of carbon emission performance from dynamic and static perspectives. The article also analyzes the global uncertainty and sensitivity. It puts forward specific measures to improve the carbon emission performance of different regions, conducive to promoting the national unified carbon trading market. (2) Calculate the size and variation trend of inter-regional and intra-regional differences in carbon emission performance of eight economic regions, which is conducive to improving carbon emission reduction targets with regional differences. (3) According to the Tobit regression model, the influencing factors of carbon emission performance values in different regions and their influencing degrees are analyzed at a deeper level, conducive to putting forward targeted suggestions for improving carbon emission performance in different regions.

2.In methodology, authors are recommended to create a study area map including the eight economic zones to let the readers get an idea about the study area.

Response: Thank you very much for your suggestions. 

A map to indicate the locations of the eight economic regions has been added, as shown in the revised paper Fig1.

3.It is also important to present some of the socio-economic and land use and environmental factor data such as per capita GDP, population density, forest area, green space, rate of urbanization, and average rainfall at a regional scale to compare the differences among the selected economic zones and compare with the study results.

Response: Thank you very much for your suggestions. 

In the "Study area " part of the revised paper, this paper adds the relevant data of socio-economic, land use, and environmental factors of the eight economic regions in China and makes a comparative analysis with the above data in the following discussion and conclusion. The added parts are as follows:

Economic indicators, land use, and environmental factors vary significantly from region to region. According to the mean value of the research period, the region with the highest economic level is ECER, which is 828005 billion yuan; the region with the lowest economic level is NWER, which is only 1,142.929 billion yuan; and the region with the highest added value of the tertiary industry is ECER, which is 3,622.941 billion yuan. The area with the most significant population density is ERMRYR, up to 14,477.85 people/square kilometers, and the region with the largest afforestation area is also ERMRYR, up to 1,459.29 thousand hectares. This is related to the characteristics of the Yellow River Basin, which is caused by massive afforestation to prevent soil erosion in this region. The region with the highest water resources per capita, NWER, is much higher than other regions, closely related to the small population in this region. The region with the highest level of urbanization is SCER (Table 1). Regional resource endowments and different development stages are the fundamental reasons for various carbon emission performances.

4.The author could reconsider the presentation of results in tabular form. It would better if the author makes some maps like Figure 2 instead of Table 4,5, & 6 for better understanding at a spatial scale. In that case, the author can replace the tables in the supplementary information section.

Response: Thank you very much for your suggestions. 

In order to analyze the results more intuitively, some of the research results are presented in the form of charts instead of tables. Table 4 in the original paper is represented by new Table 4, Fig 2 and Fig 3 in the revised paper; Table 5 in the original paper is replaced by Fig 4 in the revised paper; Table 6 in the original paper is replaced by Fig 5-6 in the revised paper.

Dear experts, because the chart is too large, it is not presented here. Please review the specific changes to the original text. Thank you very much.

The picture is too large to list them all，partly of the newly added Figures as shown below:

Fig 4 The ML index and its decomposition results.

Fig 5 Decomposition results of the Theil index in 2005-2017.

Fig 6 Inter-regional and intra-regional differences in China's eight economic regions.

5.The study failed to explore the influencing factors of regional carbon emission performance. This needs to be done, as many studies are available in the literature, and relevant data are also readily available in China as per the reviewer's understanding.

Response: Thank you very much for your suggestions. 

In this paper, the research on the influencing factors and influence degree of regional carbon emission performance is added. Based on relevant domestic and foreign kinds of literature, the Tobit model is used to select five indicators for research, and the positive and negative impacts of relevant factors and influence degree are obtained. The specific content is pointed out in " Influencing factors of carbon emission performance based on Tobit model. "And the corresponding analysis results are presented in the "Discussion."

6.It would be great if the author adds some discussion on the reasons for spatial heterogeneity in terms of carbon emission performance in the respective sections.

Response: Thank you very much for your suggestions. 

In this paper, the differences in carbon emission performance in different regions can be directly observed by drawing regional maps with data added. Through Fig 3 and Fig 7, the reasons for spatial heterogeneity are discussed in "Analysis on the spatial pattern angle of regional decomposition" "Spatial differences of regional Theil index," respectively. And in the next step of research, we will conduct a detailed discussion and research on the spatial heterogeneity of regional carbon emissions.

7.The author can also add some literature in the policy implication section especially some of the suggestions highlighted in this study that might have connections with earlier studies of similar research fields, which will improve the weightage of the suggestions being recommended in this study.

Response: Thank you very much for your suggestions. 

Considering the overall layout of the revised paper, the revised paper integrates the conclusions and policy recommendations and compares them with the relevant literature in the research field, which improves the credibility of the research. The integrated part is shown in the "Conclusion" part of the revised paper. The conclusion and policy recommendation part combines the following latest literature and compares the conclusions with them, which makes the paper more convincing.

Ou G, Xu C, Analysis of Freight Transport Carbon Emission Efficiency in Beijing-Tianjin-Hebei: A Study Based on Super-efficiency SBM Model and ML Index. Journal of Beijing Jiaotong University (Social Sciences Edition). 2020,19(02):48-57. 

Lu Y, Fang S. Analysis of Spatio-temporal evolution and influencing factors of eco-efficiency of urban construction land in Wuhan city circle based on SBM-DEA and Malmquist Model. Resources and Environment in the Yangtze Basin. 2017,26(10):1575-1586. 

Liu X, Yang X, Guo R. Regional Differences in Fossil Energy-Related Carbon Emissions in China's Eight Economic Regions: Based on the Theil Index and PLS-VIP Method. Sustainability, 2020, 12.

Sun X, Wang G, Dong H, Zhang H. Research on Efficiency of Carbon Emission of Resource-based Cities Based on DEA Model and SE- SBM Model. Science and Technology Management Research. 2016,36(23):78-84.

8.The authors didn't mention any limitations of the study. However, several limitations exist in the study. So, the authors are recommended to mention the potential limitation of the study in the discussion or conclusion section.

Response: Thank you very much for your suggestions. 

According to the research of this paper, the main advantages and limitations of this paper are summarized in the last " Strengths and limitations" part of the paper. The details are as follows:

Strengths and limitations

In this paper, the research objects are divided into eight economic regions in China, changing the research direction of provincial or industrial level in the previous carbon emission performance measurement. The research results are more targeted and more conducive to the unification of the national carbon trading market. First of all, the combination of the super-efficiency SBM model and Malmquist index model illustrates the characteristics of carbon emission performance from both static and dynamic perspectives, which makes up for the shortcomings of preliminary discussion of a single model. Secondly, through the decomposition of regional carbon emission performance differences, the inter-regional and intra-regional differences of carbon emission performance can be obtained, which provides convenience for reducing regional differences.

 However, there are still some shortcomings in this paper. This paper only considers eight primary energy consumption to estimate the carbon emissions, which has a particular gap with the actual regional carbon emissions. In the analysis of driving factors, this paper only considers the influence of five major factors on regional carbon emission performance, without in-depth study on the impact of scientific and technological progress, government macro-regulation, average rainfall, and population aging carbon emission. We will continue to pay attention to the development of regional carbon emission performance in future work. We will improve the above deficiencies to obtain a more accurate carbon emission performance value and provide suggestions for improving regional carbon emission performance.

Finally, we thank the reviewers for their valuable comments and suggestions and hope that these modifications can meet their expectations. If not, please do not hesitate to inform us. We are incredibly pleased to modify our paper according to your comments. In addition, many appreciations should be given to the academic Editor and Reviewer again for their warm work earnestly and instructive comments.

---

## [Decision Letter · Decision Letter 1]

19 Apr 2021

Analysis of carbon emission performance and regional differences in China's eight economic regions ： Based on the super-efficiency SBM model and the Theil Index

PONE-D-21-02089R1

Dear Dr. Yu,

We’re pleased to inform you that your manuscript has been judged scientifically suitable for publication and will be formally accepted for publication once it meets all outstanding technical requirements.

Kind regards,

Ghaffar Ali, PhD

Academic Editor

PLOS ONE

Additional Editor Comments (optional):

Reviewers' comments:

Reviewer's Responses to Questions

**Comments to the Author**

1. If the authors have adequately addressed your comments raised in a previous round of review and you feel that this manuscript is now acceptable for publication, you may indicate that here to bypass the “Comments to the Author” section, enter your conflict of interest statement in the “Confidential to Editor” section, and submit your "Accept" recommendation.

Reviewer #1: All comments have been addressed

Reviewer #2: All comments have been addressed

2. Is the manuscript technically sound, and do the data support the conclusions?

Reviewer #1: Yes

Reviewer #2: Yes

3. Has the statistical analysis been performed appropriately and rigorously? 

Reviewer #1: Yes

Reviewer #2: Yes

4. Have the authors made all data underlying the findings in their manuscript fully available?

Reviewer #1: Yes

Reviewer #2: Yes

5. Is the manuscript presented in an intelligible fashion and written in standard English?

Reviewer #1: Yes

Reviewer #2: Yes

6. Review Comments to the Author

Reviewer #1: Authors have addressed all the comments and I am now happy to accept the manuscript for publication.

Author could consider to rephrase the sentence on Line 72-74. it could be states as China promised to gradually reduce emission after 2030, just a suggestion.

Reviewer #2: Thanks for considering all the comments and suggestions. The manuscript's accuracy significantly improved after correction. It should be accepted for publication in its current state. Congratulations!

7. PLOS authors have the option to publish the peer review history of their article (what does this mean?). If published, this will include your full peer review and any attached files.

Reviewer #1: No

Reviewer #2: No

---

## [Editor Report · Acceptance letter]

22 Apr 2021

PONE-D-21-02089R1 

Analysis of carbon emission performance and regional differences in China's eight economic regions：Based on the super-efficiency SBM model and the Theil index 

Dear Dr. Yu:

I'm pleased to inform you that your manuscript has been deemed suitable for publication in PLOS ONE. Congratulations! Your manuscript is now with our production department. 

Kind regards, 

on behalf of

Dr. Ghaffar Ali 

Academic Editor

PLOS ONE